# The Effects of Cultivating Tobacco and Supplying Nitrogenous Fertilizers on Micronutrients Extractability in Loamy Sand and Sandy Soils

**DOI:** 10.3390/plants10081597

**Published:** 2021-08-04

**Authors:** Jacob B. Lisuma, Ernest R. Mbega, Patrick A. Ndakidemi

**Affiliations:** 1Department of Sustainable Agriculture and Biodiversity Ecosystem Management, School of Life Science and Bio-Engineering, The Nelson Mandela African Institution of Science and Technology (NM-AIST), Arusha 23000, Tanzania; ernest.mbega@nm-aist.ac.tz (E.R.M.); patrick.ndakidemi@nm-aist.ac.tz (P.A.N.); 2Department of Research, Tobacco Research Institute of Tanzania (TORITA), Tabora 45000, Tanzania

**Keywords:** micronutrients, NPK + CAN fertilizer, soil pH, tobacco nicotine

## Abstract

This research was conducted to evaluate the trends of the extractable micronutrients boron (B), copper (Cu), iron (Fe), manganese (Mn), and zinc (Zn) in soils differing in textures and collected before tobacco cultivation, and in after unfertilized and fertilized (N_10_P_18_K_24_ and CAN 27%) plots. The soils and tobacco leaves were assessed on the contents of the micronutrients after unfertilized and fertilized tobacco cultivation. In soils, tobacco cultivation with fertilization increased the extractable Cu, Fe, Mn, and Zn by 0.10, 11.03, 8.86, and 0.08 mg kg^−1^, respectively, but decreased the extractable B by 0.04 mg kg^−1^. The effects of fertilization increased the extractable Cu, Fe, Mn, and Zn by 0.14, 14.29, 9.83, and 0.24 mg kg^−1^, respectively, but decreased B by 0.08 mg kg^−1^. The combination effects of tobacco cultivation and fertilization increased the extractable Cu, Fe, Mn, and Zn by 0.24, 25.32, 18.69, and 0.32 mg kg^−1^, respectively, but decreased the extractable B by 0.12 mg kg^−1^. The results revealed that the solubility of the extractable Zn, Mn, Cu, and Fe in soils were increased by both tobacco and fertilization, but the extractable B was decreased. The fertilization of the studied soils with NPK + CAN fertilizers significantly increased the concentration of the extractable micronutrients in tobacco leaves. Based on the findings of this study, further research must be conducted to investigate the effects of tobacco cultivation on soil health and fertility beyond considering only soil pH, SOC, micronutrients, and macronutrients. These studies should include the relationship between soil fertility (pH, texture, CEC, base saturation, etc.), micronutrients, and agronomic practices on the effect of tobacco cultivation on the extractability of B, Cu, Fe, Mn, and Zn.

## 1. Introduction

The tobacco plant (*Nicotiana tabacum* L.) is the major cash plant crop grown globally in areas where food crops like cereals and legumes are also grown. The genus *Nicotiana* was coined after Jean Nicot, when it started being cultivated as a decorative plant in the years 1530–1604 [1]. The tobacco plant gained popularity with time due to its curative properties against worms, toothaches, and mitigating obesity [2]. The crop is currently used mainly for chewing, snuffing, and smoking [3] and is an important source of income for the countries producing tobacco. However, tobacco nicotine through smoking is widely known to cause cancer [4,5]. Tobacco nicotine a major metabolite by 96% in the tobacco plant, also known to be released in soils through tobacco roots, and its dynamic is highly dependent on the soil moisture and rooting depth [6]. The released nicotine is adsorbed more in acidic soils and reported to affect the soil bacteria, growth, and yield of cereal crops such as maize [7]. Nicotine in soil reduces the levels of nutrients such as phosphorus (P), potassium (K), sulfur (S), and magnesium (Mg) but it increases the levels of nitrogen (N), calcium (Ca), copper (Cu), iron (Fe), and zinc (Zn) [8,9,10].

The micronutrients including boron (B), manganese (Mn), molybdenum (Mo), nickel (Ni), Cu, Fe, and Zn are required in small quantities by the plants, but the importance of any of these nutrients cannot be replaced by any other nutrient [11]. These micronutrients are among the essential nutrients for the production of a quality tobacco leaf [11,12,13,14]. The micronutrients are essential in the protein metabolism, chlorophyll formation, and alkaloid production of tobacco [12,13,14,15,16,17,18,19,20]. However, no study has documented the effects of nutrients B and Mn in tobacco-cultivated soils. There are also no studies of the mechanisms by which tobacco nicotine influences the solubility of micronutrients in soils. The soil organic matter (SOM) and specifically soil organic carbon (SOC) in soil converts adsorbed fractions of micronutrients to more accessible forms by the plant. The SOM is composed of carbon (C), hydrogen (H), oxygen (O), nitrogen (N), phosphorus (P), and sulfur (S), which creates difficulties in measuring the actual content of SOM in soils [7]. Most analytical methods determine the content of SOC and estimate the amount of SOM through a conversion factor (SOC (%) × 1.72), which assumes that the SOM contains 58 % SOC. The SOM increases the uptake of micronutrients by the plants through increase in their solubility and the resultant nature of exchangeable forms [7]. Whereas a study conducted by Lisuma et al. [10] described the effect of nicotine released by tobacco on the availability of the extractable macronutrients in the sandy soils, the present study addressed the effect of the same nicotine on the solubility of the extractable micronutrients in the same soils. The two studies used the same experimental design and the soils for evaluating the levels of macronutrients and micronutrients were collected from the same plots. However, the methods by which the micronutrients were measured did not follow the same laboratory procedures as those used to measure the macronutrients. In the present study, only the micronutrients B, Fe, Cu, Mn, and Zn were evaluated in soils where tobacco was cultivated with the application of NPK + CAN fertilizers. 

Therefore, the objective of this research was to evaluate the effects of the tobacco (nicotine) plant on the levels of the extractable B, Cu, Fe, Mn, and Zn in soils on its own and in comparison with the application of N-containing fertilizers. The results of this study will assist tobacco growers and other stakeholders in making the appropriate decision in specifying the land/fields for tobacco cultivation and/or the methods of cultivating the subsequent food crops after the tobacco has been harvested, as the levels of micronutrients will be known.

## 2. Results

### 2.1. Soil Properties before the Establishment of Trials

The properties of soils from the Sikonge and Urambo sites were medium in reaction with pH of 5.89 and 5.87, respectively, and both sites had loamy sand soil. The Tabora site had sandy soil and was strongly acid (pH 5.47) in reaction [21]. 

### 2.2. The Effect of Tobacco Cultivation and Fertilizer Application on Soil Micronutrients

Table 1 presents the results of the effects of tobacco cultivation and fertilizer application on the extractable micronutrients before and after the trials. The solubility of the extractable B (0.33 mg kg^−1^) in the Sikonge soil was significantly higher (*P ≤* 0.001) compared with the B recorded in the Urambo (0.28 mg kg^−1^) and Tabora (0.22 mg kg^−1^) soils. The amount of the extractable B decreased significantly (*P <* 0.001) from 0.32 mg kg^−1^ before tobacco cultivation to 0.28 mg kg^−1^ in unfertilized tobacco soils, demonstrating the effect of tobacco nicotine on decreasing the extractable B by 0.04 mg kg^−1^. Furthermore, the amount of extractable B decreased further to 0.24 mg kg^−1^ in fertilized tobacco soils, indicating that the effect of fertilization resulted in a decrease in the extractable B in soils by 0.08 mg kg^−1^. In combination, tobacco nicotine and fertilizer application caused a decrease in the extractable B in soils by 0.12 mg kg^−1^. The interaction effects of sites and the fertilization of tobacco were significant (Figure 1). The solubility of the extractable B in the Tabora and Urambo soils were reduced in unfertilized and fertilized tobacco-cultivated soils. Surprisingly, the extractable B increased significantly from 0.33 to 0.37 mg kg^−1^ and decreased to 0.30 mg kg^−1^ in unfertilized and fertilized tobacco-cultivated soils, respectively, in the Sikonge site.

The solubility of the extractable Cu in soils was significantly (*P <* 0.001) high in the Tabora (0.31 mg kg^−1^) site followed by the Urambo and Sikonge sites with the extractable Cu of 0.29 and 0.24 mg kg^−1^, respectively. The extractable Cu increased significantly (*P <* 0.001) from 0.20 mg kg^−1^ before tobacco to 0.30 mg kg^−1^ after the cultivation of tobacco in unfertilized soils, indicating that the tobacco increased the quantities of the extractable Cu by 0.10 mg kg^−1^. Further, the extractable Cu increased to 0.34 mg kg^−1^ in tobacco-cultivated soils, indicating that fertilization caused an increase of the extractable Cu in soils by 0.14 mg kg^−1^. In combination, the effects of tobacco and fertilization caused the extractable Cu in soils to increase by 0.24 mg kg^−1^. The interactions of sites and fertilization were significant in the amounts of the extractable Cu in soils (Figure 2). Unlike the Sikonge site, the extractable Cu in the Tabora and Urambo sites increased significantly in unfertilized and fertilized tobacco-cultivated soils. 

The solubility of the extractable Fe in the Sikonge and Tabora sites was significantly (*P <* 0.001) higher than in Urambo site. The amounts of the extractable Fe in soils were 22.43, 22.31, and 21.38 mg kg^−1^ for the Sikonge, Tabora, and Urambo sites, respectively. The results indicated that the solubility of the extractable Fe in the studied soils increased significantly from 13.60 to 24.63 mg kg^−1^ in the unfertilized tobacco-cultivated soils, which is an increase of 11.03 mg kg^−1^ due to tobacco effect. In fertilized tobacco cultivated-soils, the levels of the extractable Fe in soil increased to 27.89 mg kg^−1^, which is an increase of the extractable Fe by 14.29 mg kg^−1^ as an effect of fertilization (Table 1). The combination of the tobacco and fertilization effects caused an increase of the extractable Fe in soils by 25.32 mg kg^−1^. The interactions of the sites and fertilization were significant on the amounts of the extractable Fe in soils. Across the sites, the levels of the extractable Fe increased significantly after tobacco cultivation in fertilized and unfertilized soils (Figure 3).

In all experimental sites, the levels of the extractable Mn in soil were significantly (*P <* 0.001) different (Table 1). The solubility of the extractable Mn increased significantly (*P <* 0.001) from 20.10 to 29.24 mg kg^−1^ in unfertilized tobacco-cultivated soils, indicating an 8.86 mg kg^−1^ increase as an effect of tobacco nicotine. Further, the amounts of the extractable Mn increased significantly (*P <* 0.001) to 29.93 mg kg^−1^ in fertilized tobacco-cultivated soils, which is an increase of the extractable Mn by 9.83 mg kg^−1^ as an effect of fertilization. The combination of tobacco and fertilization resulted in an increase of the extractable Mn in soils by 18.69 mg kg^−1^. The interactions of sites and fertilization were significant on the extractable Mn in soils (Figure 4). The extractable Mn levels in all sites increased significantly in unfertilized tobacco-cultivated soils by 7.33, 9.70, and 10.40 mg kg^−1^ in the Sikonge, Tabora, and Urambo sites, respectively. A significant increase in the extractable Mn in soils was observed in fertilized tobacco-cultivated soils in the Tabora site but not in the Sikonge and Urambo sites. 

The solubility of the extractable Zn in soils for the Urambo and Tabora sites was significantly (*P <* 0.001) lower than in the Sikonge site. The Zinc levels in soils increased significantly from 0.32 to 0.40 mg kg^−1^ in unfertilized tobacco-cultivated soils, which is an increase of the extractable Zn by 0.08 mg kg^−1^ caused by the effect of tobacco nicotine. The cultivation of tobacco in fertilized soils resulted in the increase of the extractable Zn in soils to 0.56 mg kg^−1^, which is a 0.24 mg kg^−1^ increase in the extractable Zn caused by the effect of fertilization (Table 1). The combination effects of tobacco nicotine and fertilization resulted in an increase of the extractable Zn in soils by 0.32 mg kg^−1^. The levels of the extractable Zn in soils were significantly affected by the interactions of sites and fertilization (Figure 5). In the Urambo and Sikonge sites, the levels of the extractable Zn in soils increased significantly in fertilized tobacco-cultivated soils. In the Tabora site, the levels of the extractable Zn in soils increased significantly in unfertilized and fertilized tobacco-cultivated soils.

### 2.3. The Effect of Fertilization with NPK and CAN on Micronutrient Concentrations in Tobacco Leaves 

The leaf concentrations of the extractable micronutrients (B, Cu, Fe, Mn, and Zn) in tobacco are presented in Table 2. The leaf concentration of the extractable Mn differed significantly (*P <* 0.001) across the sites. The leaf extractable B (15.58 mg kg^−1^) of tobacco cultivated in the Sikonge site was significantly (*P <* 0.001) higher than those in the Urambo (14.60 mg kg^−1^) and Tabora (13.93 mg kg^−1^) sites. Tobacco cultivated in the Urambo site had a leaf extractable Cu concentration of 12.03 mg kg^−1^ which was significantly (*P <* 0.001) higher than those recorded in leaves of tobacco cultivated in Sikonge and Tabora, each with the extractable Cu of 8.71 mg kg^−1^. Tobacco cultivated in the Tabora site had leaf extractable Fe (232.40 mg kg^−1^) and Zn (17.94 mg kg^−1^) which were significantly (*P <* 0.001) higher than those recorded in the Sikonge (Fe 139.68; Zn 13.64 mg kg^−1^) and Urambo (Fe 139.27; Fe 13.53 mg kg^−1^) sites. In the comparison of leaf concentrations of fertilized and unfertilized tobacco, the results indicated that the extractable B, Cu, Mn, and Zn increased significantly (*P <* 0.001) upon the fertilization of the tobacco plants. There was no significant increase in leaf extractable Fe concentration for unfertilized and fertilized tobacco. Furthermore, there were significant interaction effects of the sites and fertilization of the tobacco for leaf extractable Cu (*P <* 0.001) and Mn (*P =* 0.05). However, there were no significant interaction effects of the sites and fertilization on leaf concentrations of the extractable B, Fe, and Zn.

### 2.4. The Relationship between Nicotine Contents in Soil and Micronutrients

The results of the multiple linear regression analysis showing the relationship between the nicotine contents in soil and micronutrients are presented in Table 3. The nicotine in soils was regressed as a response unit (Y), while micronutrients, soil OC, and pH explanatory variables depicted a model (1): Nicotine (Y) = 95.42 + 95.58B + 64.92Cu + 41.12OC + 0.47Fe − 25.89SoilpH − 9.34Zn − 0.20Mn(1)

The coefficient of determination (R^2^) was 96%. 

The model (1) narrates that B, Cu, Fe, and SOC are positively significant at (*P* < 0.001) in influencing nicotine contents in soils. The model indicates that Mn^2+^, Zn^2+^, and soil pH negatively influenced the nicotine in soils. However, nicotine and soil Cu, Fe, Mn, and Zn showed significantly (*P* < 0.001) positive correlations (Table 4). Negative correlations were observed between nicotine in soils with SOC, soil pH, and the extractable B. The correlation results were consistent with the observed trends of these micronutrients in the studied soils.

## 3. Discussion

The effects of tobacco cultivation on soil micronutrients, which could probably be due to the nicotine effect, caused increases in the solubility of the extractable Cu, Fe, Mn, and Zn but decreases in the solubility of the extractable B. Nicotine is associated with a decrease in soil pH with an increase in acidity, which could be the reason for the observed trends of these micronutrients. The effects of fertilization with NPK + CAN fertilizers on the solubility of extractable micronutrients in soils followed a similar trend as that of the effect observed from tobacco nicotine. However, the increases in the solubility of the extractable Cu, Fe, Mn, and Zn and a decrease in solubility of the extractable B were higher than those trends observed in unfertilized tobacco-cultivated soils, suggesting that fertilization increases the impact of tobacco nicotine on the solubility of the studied micronutrients in soils. Further, the combined effects of tobacco cultivation and fertilization with NPK + CAN fertilizers indicated a greater impact of these factors on the solubility of the studied micronutrients in soils. The extractable B decrease in soils could be related to its high absorption by the tobacco plants due to its essential role. The extractable B from soils is reported to improve the sugars, nicotine, organic acids, and amino acids content in tobacco plants [12,23,24]. The NPK and CAN fertilizers and tobacco crop influence on the solubility of extractable Cu is in line with other studies which have demonstrated the influence of N, P, K, and Ca nutrients applied as NPK/CAN fertilizer on the solubility of the extractable Cu in soils [24,25,26].

The levels of the extractable Fe in soils before tobacco were high, but its solubility was very low. The increase in soil acidity by the tobacco crop also increased the solubility of the extractable Fe in soils [8]. The tobacco crop demonstrates the most considerable influence on increasing the solubility of the extractable Mn in the studied soils due to its role in enhancing chlorophyll formation and resistance to diseases [17,18]. Other studies also reported the uptake of manganese by plants to be high [27,28,29]. The present study also indicated that NPK and CAN had an enormous influence on the increases of the extractable Zn solubility in the studied soils, similar to the observations reported in other related studies [8,22]. Tobacco plants release nicotine into the soil environment, which is also acidic, and hence increase the solubility of many micronutrients, thereby making them readily available to the tobacco plant [10]. The trends of these micronutrients in soils will also have varying impacts on the overall soil health and fertility and may affect the yields of the subsequent crops [8,9,10]. Further research is required to investigate the effects of tobacco cultivation on the overall soil health and fertility beyond considering only soil pH, SOC, micronutrients, and macronutrients. Some prominent studies can investigate the relationship between soil fertility (pH, texture, CEC, base saturation, etc.), micronutrients, and agronomic practices on the effect of tobacco cultivation on the availability of the extractable B, Cu, Fe, Mn, and Zn. The present study showed a high relationship between nicotine secreted by the tobacco plant and the diethylene triamine pentaacetic acid (DTPA)-extracted micronutrients in soils. The nicotine secreted by the tobacco plants into the soil system lowered the soil pH (increased acidity), which in turn increased the availability of the micronutrients extracted from tobacco-cultivated soils and/or the tobacco leaves. Similar findings were reported by Golia et al. [30] when studying the relationship between tobacco nicotine and DTPA–extractable micronutrient levels.

The fertilization of tobacco with NPK + CAN fertilizers increased the leaf concentrations of the extractable B, Cu, Fe, Mn, and Zn. The tobacco plants cultivated in soils of the Sikonge site had significantly (*P* < 0.001) higher concentrations of the leaf extractable B than the leaf B of tobacco plants cultivated in the Urambo and Tabora sites, which reflect higher leaf nicotine and yield (Tables S4–S7 deposited in [10]. The tobacco plants cultivated in soils at the Urambo sites had significantly (*P* < 0.001) greater concentrations of leaf Cu than those of the Tabora and Sikonge sites. The Tabora site recorded significantly (*P* < 0.001) higher leaf concentrations of Fe, Mn, and Zn than those in the Sikonge and Urambo sites. These differences could be attributed to the low contents of these nutrients in the soils. The results indicate that the leaf concentrations of the extractable B in tobacco cultivated in the Tabora, Urambo, and Sikonge soils ranged from 13.93 to 15.58 mg kg^−1^. Unfertilized and fertilized tobacco plants had leaf B concentrations of 14.08 and 15.33 mg kg^−1^, respectively, which were within the established sufficient B range of 14 to 50 mg kg^−1^ [25]. The leaf concentrations of Cu in fertilized and unfertilized tobacco were sufficient and insufficient, respectively, based on the critical range of 10 to 60 mg kg^−1^ [25]. Based on the established critical values for Fe in tobacco leaves (50–200 mg kg^−1^), both fertilized and unfertilized tobacco had sufficient leaf Fe concentrations [15,25]. The leaf extractable Mn concentrations in fertilized and unfertilized tobacco were within the sufficient range of 26–400 mg kg^−1^ [25]. The leaf extractable Zn concentrations in fertilized and unfertilized tobacco plants were below the critical range of 17–110 mg kg^−1^ [25]. According to Zeng et al. [31], the concentrations of the extractable Zn, Mn, and Fe in tobacco leaves increased with increasing soil acidity (lowering of pH), but the increase in the extractable Cu concentration was no significant. Other researchers have observed the inconsistent relationship between the extractable B and other micronutrients in soils. Ali et al. [32] found that extractable B in soils caused an increase in the solubility of the extractable Zn, decreased the solubility of the extractable Cu and Mn, but had no effect on the extractable Fe. 

### Conclusions

The nicotine from tobacco and the use of NPK + CAN fertilization in soils where tobacco was cultivated increased the solubility of the extractable Cu, Fe, Mn, and Zn, but had a decreasing effect on the solubility of B. The trend by which these micronutrients were increased was tobacco + fertilization > fertilization > tobacco. The increase of micronutrients due to fertilization provided an insight that these nutrients had a positive relationship with the applied N, P, K, and Ca from NPK + CAN fertilizer. The decreased solubility of the extractable B explained the contrast in soils. Fertilization also significantly increased the concentration of these micronutrients in tobacco leaves, indicating their importance to tobacco crops. Further research must be conducted to investigate the effects of tobacco cultivation on soil health and fertility beyond considering only soil pH, SOC, micronutrients, and macronutrients. Future studies should include the relationship between soil fertility (pH, texture, CEC, base saturation, etc.), micronutrients, and agronomic practices on the effect of tobacco cultivation on the availability of the extractable B, Cu, Fe, Mn, and Zn.

## 4. Materials and Methods

### 4.1. Description of the Study Areas and Location 

The experiment was conducted during the 2017–2018 cropping season in three major tobacco-producing districts, Urambo, Sikonge and Tabora, in the Tabora region, Tanzania. Fallow sites with no tobacco background were selected. The criteria for site selection were based on the initial survey conducted in different tobacco cultivating parts of the country. The Urambo district had 890 mm and 25 °C mean rainfall and temperature, respectively, with coordinates 05°31′47.4″ S, 032°50′03.2″ E; 1191 m a.s.l. The Sikonge district had 1050 mm and 29 °C mean rainfall and temperature, respectively, with coordinates 05°04′33.5″ S, 032°00′09.8″ E; 1108 m a.s.l. The Tabora district had 950 mm and 27 °C mean rainfall and temperature, respectively, and is located at 05°03′44.4″ S, 032°40′07.4″ E; 1160 m a.s.l. Based on the soil texture of the soil properties [22], the SOC was very low (see Tables S1–S3) deposited in [10]. Referring to our previous studies [7,10], there is no background of soil classification in the Tabora region using the United States Department of Agriculture (USDA) soil taxonomy and/or the World Reference Base (WRB-FAO). Therefore, it was not possible to know the spatial variability of the background ambient values of Fe, Mn, and other trace elements that are due to combinations of geogenic and anthropogenic sources. Thus, this is left as an area that requires further investigation in this tobacco-cultivating region.

### 4.2. Experimental Design, Treatments and Experimentation 

A randomized complete block design (RCBD) was used with treatments replicated three times. The factors and their treatments in brackets were: - (1) sites (Tabora, Urambo, Sikonge); and (2) tobacco-cultivation regimes (unfertilized, fertilized). In evaluating the effects of tobacco nicotine and fertilizers, the element of soils before tobacco cultivation was included. The fertilizers used were N_10%_P_18%_K_24%_ blended with 0.012% B, 3% CaO, 0.5% MgO, 7% S; and calcium ammonium nitrate (CAN 27N%) blended with 14% NO_3_, 13% NH_4_, 3% CaO, 1.7% MgO, 3% S, as reported in a parallel study by Lisuma et al. [10]. The varieties of tobacco, field preparation, plot size, and sowing are as described by Lisuma et al. [10].

### 4.3. Data Collection from Soil and Tobacco Plants 

The diethylene triamine pentaacetic acid extraction method of B, Cu, Fe, Mn, and Zn was deployed in this study due to the limitation of other methods [33]. The selection of the DTPA method for the extraction of the studied micronutrients was also based on its non-equilibrium property as it estimates the potential soil availability of the extractable B, Fe, Cu, Zn, and Mn. Further, this method offered the most favorable combination of stability constants for the simultaneous determination of all the five micronutrients (B, Fe, Cu, Zn, and Mn) and it avoided their dissolution when occluded on it [34]. According to Malathi and Stalin [34], DTPA also chelates and provides the availability or toxicity indices of B, Zn, Cu, Mn, and Fe to plants. The pH, SOC, and nicotine in the same soils and plants (Tables S1–S3) deposited [10] were determined as described by Lisuma et al. [10]. 

### 4.4. Statistical Analyses

The analysis of variance (ANOVA) using two factors: (i) site (Sikonge, Tabora and Urambo); and (ii) treatments (soil before tobacco cultivation, soil after unfertilized and soil after fertilized tobacco cultivation) were performed using STATISTICA, 8th Edition. The significant means were compared with the Fisher Least Significance Difference at *P* = 0.05. Nicotine was regressed as a response variable (Y) while keeping soil B, Cu, Fe, Mn, and Zn as explanatory variables following the model (2): Nicotine (Y) = m_1_X_i_ + m_2_X_ii_ + m_3_X_iii_ + m_4_X_iv_ + m_5_X_v_ + m_6_X_vi_ + m_7_X_vii_ + C(2)
where X_i_ to X_vii_ stand for the parameters B, Cu, Fe, Mn, Zn, SOC, and Soil pH; m_1_ to m_7_ represent coefficients of the parameters; and C is the constant. 

Nicotine was further correlated with the B, Cu, Fe, Mn, Zn, SOC and soil pH in order to measure their relationships.

## Figures and Tables

**Figure 1 plants-10-01597-f001:**
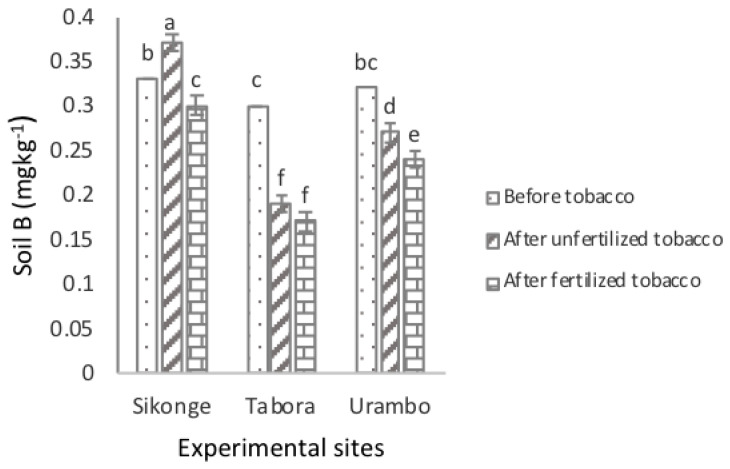
Effect of tobacco and NPK + CAN fertilization on the extractable B in soils.

**Figure 2 plants-10-01597-f002:**
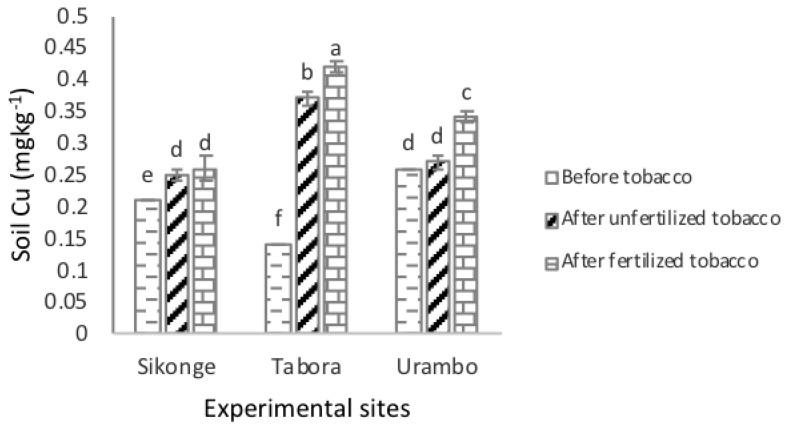
Effect of tobacco and NPK + CAN fertilization on the extractable Cu in soils.

**Figure 3 plants-10-01597-f003:**
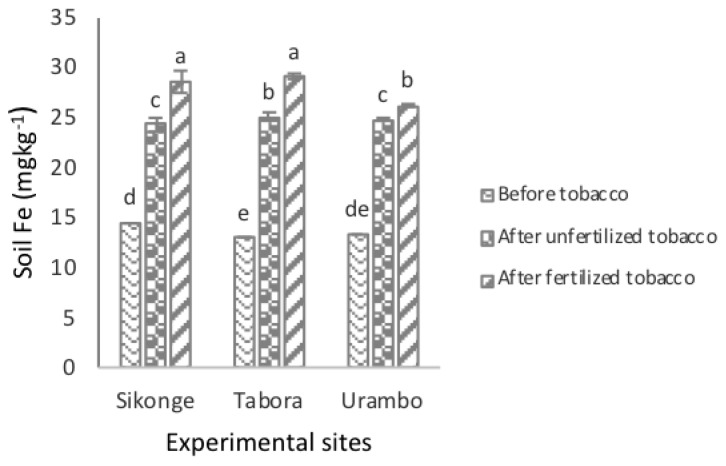
The effect of tobacco and NPK + CAN fertilization on the extractable Fe in soils.

**Figure 4 plants-10-01597-f004:**
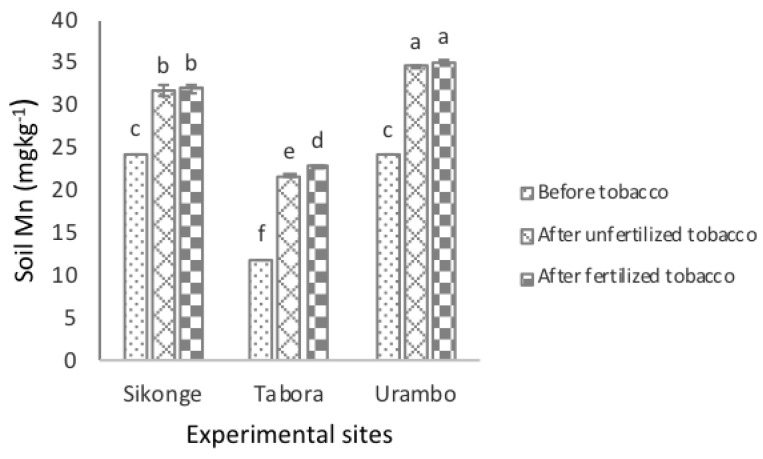
The effect of tobacco and NPK + CAN fertilization on the extractable Mn in soils.

**Figure 5 plants-10-01597-f005:**
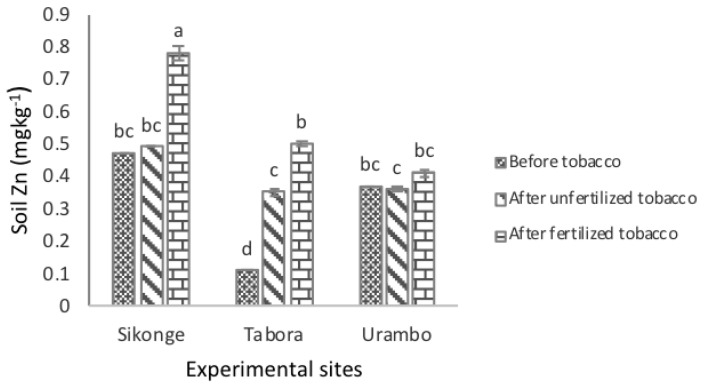
The effect of tobacco and NPK + CAN fertilization on the extractable Zn in soils.

**Table 1 plants-10-01597-t001:** Selected soil properties of the Sikonge, Tabora and Urambo experimental sites as affected by tobacco cultivation.

Site:	Boron (mg kg^−1^)	Copper (mg kg^−1^)	Iron (mg kg^−1^)	Manganese (mg kg^−1^)	Zinc (mg kg^−1^)
Sikonge	0.33 ± 0.01 a	0.24 ± 0.01 c	22.43 ± 2.10 a	29.28 ± 1.26 b	0.58 ± 0.06 a
Tabora	0.22 ± 0.02 c	0.31 ± 0.04 a	22.31 ± 2.41 a	18.79 ± 1.73 c	0.32 ± 0.06 b
Urambo	0.28 ± 0.01 b	0.29 ± 0.01 b	21.38 ± 2.03 b	31.21 ± 1.79 a	0.38 ± 0.01 b
Treatment:					
Soil before tobacco cultivation †	0.32 ± 0.00 a	0.20 ± 0.02 c	13.60 ± 0.24 c	20.10 ± 2.05 c	0.32 ± 0.05 c
Soil after tobacco–unfertilized	0.28 ± 0.02 b	0.30 ± 0.02 b	24.63 ± 0.24 b	29.24 ± 1.96 b	0.40 ± 0.00 b
Soil after tobacco–fertilized	0.24 ± 0.02 c	0.34 ± 0.02 a	27.89 ± 0.57 a	29.93 ± 1.84 a	0.56 ± 0.07 a
2-Way ANOVA F Statistic:					
Site (S)	133.93 ***	35.49 ***	4.10 *	1461.21 ***	24.77 ***
Treatment (T)	69.43 ***	118.53 ***	696.15 ***	985.93 ***	21.37 ***
S × T	19.90 ***	43.50 ***	4.51 **	12.44 ***	5.13 **

Means with similar letter(s) do not differ significantly based on their respective Least Significance Difference (LSD) value at 5% error rate; values presented are the means ± SE (Standard Error); *, **, *** indicate significant at *P* ≤ 0.05, *P* ≤ 0.001, *P* < 0.001, respectively; ns = non-significant; **†** Landon [22] ratings: B was very low (0–0.4), Cu was low/deficient (0–0.4), Fe was high ( >4.5), Mn was high ( >1.0), Zn was low/deficient (0–0.5), OC was very low (< 0.6).

**Table 2 plants-10-01597-t002:** Micronutrient concentrations of tobacco plant cultivated on fertilized relative to unfertilized soils.

	B (mg kg^−1^)	Cu (mg kg^−1^)	Fe (mg kg^−1^)	Mn (mg kg^−1^)	Zn (mg kg^−1^)
**Site**					
Sikonge	15.58 ± 0.39 a	8.71 ± 0.24 b	139.68 ± 1.17 b	102.55 ± 1.10 c	13.64 ± 0.19 b
Tabora	13.93 ± 0.32 b	8.71 ± 0.24 b	232.40 ± 0.39 a	233.36 ± 1.11 a	17.94 ± 0.26 a
Urambo	14.60 ± 0.35 b	12.03 ± 0.07 a	139.27 ± 8.96 b	220.98 ± 0.75 b	13.53 ± 0.18 b
Treatments					
Unfertilized tobacco	14.08 ± 0.27 b	9.41 ± 0.62 b	165.58 ± 16.55 a	180.17 ± 20.01 b	14.61 ± 0.74 b
Fertilized tobacco	15.33 ± 0.32 a	10.22 ± 0.49 a	175.32 ± 15.29 a	184.43 ± 19.86 a	15.45 ± 0.72 a
Two-Way ANOVA F-statistics					
Site (S)	13.09 ***	4500.60 ***	124.846 ***	74594.00 ***	457.59 ***
Treatment (T)	22.09 ***	608.50 ***	3.086ns	213.00 ***	38.31 ***
S × T	1.18ns	63.90 ***	1.335ns	4.00 *	0.00ns

*, *** significant at *P* ≤ 0.05, *P* < 0.001, respectively; ns = non-significant.

**Table 3 plants-10-01597-t003:** Multiple linear regression analysis of nicotine as a response parameter and the measured micronutrients in soil.

Fitted Parameters	Coefficients	Standard Error	T-Statistic	P-Value	Lower 95%	Upper 95%
Intercept	95.42434313	34.90534505	2.733803175	0.021055512	17.65038767	173.1982986
B (mg kg^−1^)	95.58563717	41.95593708	2.278238643	0.045923021	2.101983691	189.0692906
Cu (mg kg^−1^)	64.92308337	27.71679316	2.342373556	0.041172614	3.166219676	126.6799471
Fe (mg kg^−1^)	0.475368305	0.252771336	1.880625838	0.089431575	−0.087841329	1.038577938
Mn(mg kg^−1^)	−0.203960288	0.126413294	−1.613440181	0.13772393	−0.485626661	0.077706084
Zn (mg kg^−1^)	−9.339325774	7.857166985	−1.188637812	0.262043109	−26.8461848	8.167533251
OC (%)	41.11614161	15.0755355	2.727342032	0.02129032	7.525755258	74.70652796
Soil pH	−25.88681638	8.139113946	−3.180544781	0.00981037	−44.02189238	−7.751740375

**Table 4 plants-10-01597-t004:** Correlations between nicotine and other measured variables in soils.

Response Variable	Explanatory Variables						
	B	Cu	Fe	Mn	Zn	SOC	Soil pH
Nicotine	−0.47	0.52	0.88	0.44	0.74	−0.06	−0.7

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
