# Peer review of "The Effects of Cultivating Tobacco and Supplying Nitrogenous Fertilizers on Micronutrients Extractability in Loamy Sand and Sandy Soils"

_plants, 2021, doi:10.3390/plants10081597_

Round 1

Reviewer 1 Report

Referee Report on the Manuscript

‘Effects of Cultivating Tobacco and Supplying Nitrogenous Fertilizers on Micronutrients Availability in Loamy Sand and Sandy Soils'

(Manuscript's authors Jacob Lisuma, Ernest Mbega, Patrick Alois Ndakidemi)

This manuscript explored the effects of cultivating tobacco and supplying nitrogenous fertilizers on micronutrients availability in loamy sand and sandy soils.

It appears to have the potential for manuscript publication in Plants.

At the same time, there are the following questions, as well as comments and suggestions to this manuscript that could be used to improve the manuscript else.

  1.  

To be the manuscript more informative for the readers, the authors should provide in the introduction of a full range of information about the harmful and beneficial aspects of growing and tobacco processing for humans. To provide moral and ethical aspects it is also necessary to note health problems here in connection with the implications of active and especially secondhand smoke.

  1.  

It would be better if the authors provide the full Latin names of plants and the author of the classification in the manuscript for the research object.

3.

How does nicotine in soil affect other plants?

What about the allelopathic effects of nicotine on different plant species?

What about other allelopathic substances present in tobacco?

If, as the authors recommend, to cultivate plants after tobacco on these soils, then how much does this benefit ('increased levels of soil micronutrients') compensate for the effect of nicotine on plants or overlapping there or something else?

What about nicotine stability in the soil?

What plants do authors recommend to grow after tobacco on these soils?

  1.  

Concerning line 23 and lines 48-50:

If the authors have relevant data on the cultivation of any plants on this soil after tobacco, this would strengthen the manuscript.

5.

Lines 46-48:

It is desirable to define the concept of 'soil fertility' here before this.

Only B, Cu, Fe, Mn and Zn here?

  1.  

By the way, was the content of heavy metals (xenobiotics) determined in soils, for example, Cd, Pb, Cr, etc., before and after tobacco cultivation? The same - in plants?

What about non-essential elements?

What about the situation with soil HM pollution in Tanzania?

Is there such a problem?

  1.  

Lines 48-50:

How?

  1.  

The authors of the manuscript recently published a series of articles addressing issues related to tobacco, macronutrients, nicotine, soil - there may be some connection with the current manuscript.

It is necessary to discuss these works in connection with the current manuscript and reveal its novelty in comparison with these.

Lisuma, J., Mbega, E., & Ndakidemi, P. (2020). Influence of Tobacco Plant on Macronutrient Levels in Sandy Soils. Agronomy, 10(3), 418.

Lisuma, J. B., Mbega, E. R., & Ndakidemi, P. A. (2019). Influence of Nicotine Released in Soils to the Growth of Subsequent Maize Crop, Soil Bacteria and Fungi. INTERNATIONAL JOURNAL OF AGRICULTURE AND BIOLOGY, 22(1), 1-12.

Lisuma, J. B., Mbega, E. R., & Ndakidemi, P. A. (2019). Dynamics of nicotine across the soil–tobacco plant interface is dependent on agro-ecology, nitrogen source, and rooting depth. Rhizosphere, 12, 100175.

Others?

  1.  

Lines 67-68:

What is the cause of this phenomenon?

10.

Line 93 and lines 147-149:

But the content of other HM could also increase?

It could be a problem.

11.

The Discussion section should be strengthened by a larger comparison of the data obtained with the literature data.

12.

Lines 263-264:

But how do the authors take into account here the effect of nicotine, various allelochemicals to recommend this idea?

The authors should provide both additional information and more discussion for the results.

13.

Lines 302-303.

Tobacco seed variety K326.

If this is used for the manufacture of cigarettes, then the negative effects of smoking should be noted, as in point 1 above. Moral and ethical aspects.

14.

Line 309:

fertilizer CAN 27%

Provide the composition, please.

15.

Line 316:

Provide tobacco root architecture and length, please.

Lines 14-16:

“The results revealed that the unfertilized tobacco soils increased micronutrients concentration in the following order: Fe2+ (81%) > Cu2+ (50%) > Mn2+ (46%) > Zn2+ (25%) and decreasing B by 4%".

Perhaps it would be worth rephrasing?

Also in the abstract:

"B by 4%"

"Mn2+ (2%)"

Is this significant?

Changes and additions made in accordance with the above may create prerequisites for further improvement of the abstract and the manuscript as a whole.

For the abstract this must be done by reinforcing the topicality there, the fundamental concept of research and potential relevance for practice, as well as more clearly identifying and capturing the goal, objectives, and results in a more conceptualized form.

It would be much better if the abstract contains more generalized fundamental conclusions than "partial/common" results.

The recommendation for this manuscript suggests 'to be accepted' after Revisions.

Author Response

Response to Reviewer 1 Comments

Referee Report on the Manuscript ‘Effects of Cultivating Tobacco and Supplying Nitrogenous Fertilizers on Micronutrients Availability in Loamy Sand and Sandy Soils'

(Manuscript's authors Jacob Lisuma, Ernest Mbega, Patrick Alois Ndakidemi)

This manuscript explored the effects of cultivating tobacco and supplying nitrogenous fertilizers on micronutrients availability in loamy sand and sandy soils. It appears to have the potential for manuscript publication in Plants. At the same time, there are the following questions, as well as comments and suggestions to this manuscript that could be used to improve the manuscript else.

Point 1: To be the manuscript more informative for the readers, the authors should provide in the introduction of a full range of information about the harmful and beneficial aspects of growing and tobacco processing for humans. To provide moral and ethical aspects it is also necessary to note health problems here in connection with the implications of active and especially secondhand smoke.

Response 1: Comment responded positively by providing detailed information about the harmful effects and beneficial aspect of growing tobacco for humans. Healthy problems caused by smoking tobacco is well explained in the introduction section line 32-42 on page 1.

Point 2:It would be better if the authors provide the full Latin names of plants and the author of the classification in the manuscript for the research object

Response 2:Full Latin names of the  plant and the author coined the genus Nicotianais included in the introduction-section 1, line 29-31 on page 1.

Point 3: How does nicotine in soil affect other plants?

What about the allelopathic effects of nicotine on different plant species?

What about other allelopathic substances present in tobacco?

If, as the authors recommend, to cultivate plants after tobacco on these soils, then how much does this benefit ('increased levels of soil micronutrients') compensate for the effect of nicotine on plants or overlapping there or something else?

What about nicotine stability in the soil?

What plants do authors recommend to grow after tobacco on these soils?

Response 3: Comment responded positively how does nicotine affect other plants, nutrients, bacteria and its adsorption behaviours is well explained in section 1 of introduction, line 35-41 on page 1.

Point 4: Concerning line 23 and lines 48-50:

If the authors have relevant data on the cultivation of any plants on this soil after tobacco, this would strengthen the manuscript.

Response 4: No relevant data of any crop from the soil understudies, as experimental sites were selected with no historical background growing either tobacco or other crops. The primary objective was to get the effects of tobacco to the soil nutrients.

Point 5: Lines 46-48:

It is desirable to define the concept of 'soil fertility' here before this.

Only B, Cu, Fe, Mn and Zn here?

Response 5: Comment responded positively, the definition of soil fertility has been given in section 1 line 86-88 on page 2.

Point 6: By the way, was the content of heavy metals (xenobiotics) determined in soils, for example, Cd, Pb, Cr, etc., before and after tobacco cultivation? The same - in plants?

What about non-essential elements?

What about the situation with soil HM pollution in Tanzania?

Is there such a problem?

Response 6: Neither heavy metals (Cd, Pb, Cr etc.)  nor non-essential elements (Na, Si, Al, Sr, V etc.) were not determined. There is no incidence of soil heavy metal pollution in tobacco growing areas. However, levels of Fe and Mn are high while levels of Cu and Zn is low and B is inadequate. These details are given as footnote under Table 1 on page 4. Soil HM pollution recently reported in the city of Dar es Salaam to the industrial areas located about 1,000 km away from tobacco-growing areas.

Point 7: Lines 48-50:

How?

Response 7: Soil nutrients, nicotine levels and soil pH were determined first before growing tobacco. Then a study of the mechanism of the tobacco plant in influencing soil fertility (B, Cu, Fe, Mn and Zn) involved planting tobacco without fertilization followed. In this scenario levels of nicotine released, changes in soils acidity and micronutrients levels were determined. These data were compared with the data obtained before planting tobacco and after planting tobacco under fertilization. See Table 1 on page 4 for more details on the design of the experiments and data collected.

Point 8: The authors of the manuscript recently published a series of articles addressing issues related to tobacco, macronutrients, nicotine, soil - there may be some connection with the current manuscript.

It is necessary to discuss these works in connection with the current manuscript and reveal its novelty in comparison with these.

Lisuma, J., Mbega, E., & Ndakidemi, P. (2020). Influence of Tobacco Plant on Macronutrient Levels in Sandy Soils. Agronomy, 10(3), 418.

Lisuma, J. B., Mbega, E. R., & Ndakidemi, P. A. (2019). Influence of Nicotine Released in Soils to the Growth of Subsequent Maize Crop, Soil Bacteria and Fungi. INTERNATIONAL JOURNAL OF AGRICULTURE AND BIOLOGY, 22(1), 1-12.

Lisuma, J. B., Mbega, E. R., & Ndakidemi, P. A. (2019). Dynamics of nicotine across the soil–tobacco plant interface is dependent on agro-ecology, nitrogen source, and rooting depth. Rhizosphere, 12, 100175.

Others?

Response 8: Comments responded positively, our previous research output on dynamics of nicotine release in the soils, the influence of nicotine on macronutrients availability and influence of tobacco plant to the growth of maize and soil bacteria are linked in the manuscript on section 1 line 34-44; 87-90 on page 1-2 and on section 3 line 320-326 page 12 and line 378-380 and on section 3.1 line 390-391 on page 15.

Point 9: Lines 67-68:

What is the cause of this phenomenon?

Response 9: The cause of the B decrease in soils is related to its high absorption due to its essential role in the tobacco plants. B from the soils is reported to improve sugars, nicotine, organic acids and amino acids contents in tobacco plant These details are explained online 283-287 of page 11.

Point 10: Line 93 and lines 147-149:

But the content of other HM could also increase?

It could be a problem.

Response 10: Comment responded positively, even though HM were not determined, their contents could also increase in the soils and cause a problem to soil fertility. Changes have been done to indicate this and even recommended for further studies in section 3 line 324-325 page 12 and in section 3.1 line 397-398 on page 13.

Point 11: The Discussion section should be strengthened by a larger comparison of the data obtained with the literature data.

Response 11: Comment responded positively; the discussion section has been strengthened by comparing results of previous studies and other literature inline 320-330; 377-379; 382-384 and 395-398 on page 12-13.

Point 12: Lines 263-264:

But how do the authors take into account here the effect of nicotine, various allelochemicals to recommend this idea?

The authors should provide both additional information and more discussion for the results.

Response 12: Comment responded positively-dditional information provided inline 382-386 on page 13. In addition to this in our study, we took into account of nicotine affecting micronutrients as nicotine is a primary metabolite in tobacco by 96% in comparisons to other allelochemicals such as phenols, aromatic hydrocarbon, nitrosamines, alkyne, cadmium which exist in minute levels of about 4%.

Point 13: Lines 302-303.

Tobacco seed variety K326.

If this is used for the manufacture of cigarettes, then the negative effects of smoking should be noted, as in point 1 above. Moral and ethical aspects.

Response 13: Comment responded positively, tobacco seed variety K326 was used in the current study, and the effect of smoking is well noted in section 1 line 33-34 on page 1.

Point 14: Line 309:

fertilizer CAN 27%

Provide the composition, please.

Response 14: Comment responded positively. Composition of CAN 27%  includes 14% NO3, 13% NH4, 3%CaO, 1.7%MgO and 3%S. This composition also was provided in section 4.2 line 437-438.

Point 15: Line 316:

Provide tobacco root architecture and length, please.

Lines 14-16:

“The results revealed that the unfertilized tobacco soils increased micronutrients concentration in the following order: Fe2+ (81%) > Cu2+ (50%) > Mn2+ (46%) > Zn2+ (25%) and decreasing B by 4%".

Perhaps it would be worth rephrasing?

Also in the abstract:

"B by 4%"

"Mn2+ (2%)"

Is this significant?

Changes and additions made in accordance with the above may create prerequisites for further improvement of the abstract and the manuscript as a whole.

For the abstract this must be done by reinforcing the topicality there, the fundamental concept of research and potential relevance for practice, as well as more clearly identifying and capturing the goal, objectives, and results in a more conceptualized form.

It would be much better if the abstract contains more generalized fundamental conclusions than "partial/common" results.

The recommendation for this manuscript suggests 'to be accepted' after Revisions.

Response 15: Comment responded positively. Tobacco root architecture and length provided in section 4.2 line 443-444 on page 14.

Results for unfertilized and fertilized tobacco on Fe2+(81%) > Cu2+(50%) > Mn2+(46%) > Zn2+(25%) and decreasing B by 4%" have been revised. Details are indicated inline 14-18 of the abstract on page 1.

Reviewer 2 Report

Lisuma et al. present a study of the effect of tobacco cultivation on DTPA extractable trace elements from three soils.  They conclude, through statistical association, that nicotine release by the crop increases the availability of B, Cu, Fe, Mn, and Zn.  This association is insufficient to infer causality.  This inference could have been tested by the simple means of amending the DTPA extractant with nicotine.

In addition, the authors do not explain the reasons for the two main choices, i.e. sites and DTPA extraction.  Finally, it is respectfully suggested that the authors consider presenting the critical raw data, such as soil properties and the foliar analysis, that are essential to the story, and consider how to present the data in a way that readers can easily understand.  Currently there is too much statistical analysis and not enough concept development.

Author Response

Response to Reviewer 2 Comments

Point 1: Lisuma et al. present a study of the effect of tobacco cultivation on DTPA extractable trace elements from three soils.  They conclude, through statistical association, that nicotine release by the crop increases the availability of B, Cu, Fe, Mn, and Zn.  This association is insufficient to infer causality.  This inference could have been tested by the simple means of amending the DTPA extractant with nicotine.

In addition, the authors do not explain the reasons for the two main choices, i.e. sites and DTPA extraction.  Finally, it is respectfully suggested that the authors consider presenting the critical raw data, such as soil properties and the foliar analysis, that are essential to the story, and consider how to present the data in a way that readers can easily understand.  Currently there is too much statistical analysis and not enough concept development.

Response 1: DTPAdetermination of micronutrients were determined on both soil before tobacco production, soil after unfertilized tobacco and soil after fertilized tobacco. Therefore the effect of DTPA cut across all the treatment plots. Furthermore, this method was the only available in the country of study. With regards, to the issue of having not enough concept development, be assured that we have provided enough concept development from the introduction section inline 29-45 on page 1; line 85-90; 92-95 on page 2; Section 2 inline 223-224 on page 6; In discussion section line 283-287 on page 11; line 320-330 page 12; line 377-379; 382-386; conclusion section line 397-400 on page 13 and all sections reflected in abstract (line 14-18, 22-24);

Round 2

Reviewer 2 Report

The key issue of association being equated to causation has not been addressed.   The hypotheses that 1) DTPA extractable elements are plant available; and 2) excreted nicotine affects DTPA extraction; are unsubstantiated by the data. 

Author Response

Reviewer 2 comments

The key issue of association being equated to causation has not been addressed.   The hypotheses that 1) DTPA extractable elements are plant available; and 2) excreted nicotine affects DTPA extraction; are unsubstantiated by the data. 

Point 1:The key issue of association being equated to causation has not been addressed.   The hypotheses that 1) DTPA extractable elements are plant available

Response 1: Comment responded positively in section 4.3 line 706-712 by indicating that, “The selection of DTPA method for the extraction of the studied micronutrients is also based on its non-equilibrium property as it estimates the potential soil availability of B, Fe, Cu, Zn, and Mn. Further, this method offers the most favourable combination of stability constants for the simultaneous determination of all the five micronutrients (B, Fe, Cu, Zn, and Mn) and it avoids their dissolution when occluded on it [42]. According to Malathi and Stalin [42], the DTPA also chelates and provides availability or toxicity indices of B, Zn, Cu, Mn, and Fe to plants.”

Point 2:Excreted nicotine affects DTPA extraction; are unsubstantiated by the data. 

Response 2: Comment responded in line 516-521. The present study showed a high relationship between nicotine secreted by the tobacco plant and the DTPA-extracted micronutrients in soils. The nicotine secreted by the tobacco plants to the soil system lowered the soil pH (increased acidity), which in turn increased availability of the micronutrients extracted from tobacco cultivated soils and/or tobacco leaves. Similar findings are reported by Golia et al. [43] when studying the relationship between tobacco nicotine and DTPA–extractable micronutrient levels.

Round 3

Reviewer 2 Report

The authors have responded to the criticisms effectively, other than to:
  •  change the wording in the title to reflect the content, i.e. to replace the word 'availability' with 'extractability'; and
  • add the word 'extractable' when referring to changes in element concentrations because it is not possible for a crop to increase the concentration of an element in a soil.
If those changes are made then I recommend publication.

Author Response

Reviewer 2 Comments

Point 1:The authors have responded to the criticisms effectively, other than to:

  •  change the wording in the title to reflect the content, i.e. to replace the word 'availability' with 'extractability'; and
  • add the word 'extractable' when referring to changes in element concentrations because it is not possible for a crop to increase the concentration of an element in a soil.

If those changes are made then I recommend publication. 

Response 1: Comment responded positively by adding extractable when referring to changes in element concentrations in the followings lines 4, 12, 18, 25, 29, 73, 74, 82, 95, 97-100, 102, 103, 109, 111, 112, 113, 119, 121, 122, 130, 137-148, 150, 151, 153, 154, 155, 157, 159, 160, 162, 167, 169, 170-178, 183-197, 206-221, 248, 255, 257-261, 263, 265, 266, 270, 271, 273, 275, 276-279, 282, 291, 298, 299, 301, 305-314, 319-322, 326, 330, 336, 354, 356, 374.
